# Photon–Phonon Atomic Coherence Interaction of Nonlinear Signals in Various Phase Transitions Eu^3+^: BiPO_4_

**DOI:** 10.3390/nano12234304

**Published:** 2022-12-04

**Authors:** Huanrong Fan, Faizan Raza, Irfan Ahmed, Muhammad Imran, Faisal Nadeem, Changbiao Li, Peng Li, Yanpeng Zhang

**Affiliations:** 1Key Laboratory for Physical Electronics and Devices of the Ministry of Education & Shaanxi Key Lab of Information Photonic Technique, Xi’an Jiaotong University, Xi’an 710049, China; 2State Key Lab of Modern Optical Instrumentation, Centre for Optical and Electromagnetic Research, College of Optical Science and Engineering, International Research Center for Advanced Photonics, Zhejiang University, Hangzhou 310027, China; 3Department of Electrical Engineering, Sukkur IBA University, Sukkur 65200, Pakistan; 4Department of Physics, City University of Hong Kong, Hong Kong SAR 99907, China

**Keywords:** atomic coherence, spectral interaction, phonon/photon dressing, spontaneous four-wave mixing

## Abstract

We report photon–phonon atomic coherence (cascade- and nested-dressing) interaction from the various phase transitions of Eu^3+^: BiPO_4_ crystal. Such atomic coherence spectral interaction evolves from out-of-phase fluorescence to in-phase spontaneous four-wave mixing (SFWM) by changing the time gate. The dressing dip switch and three dressing dips of SFWM result from the strong photon–phonon destructive cross- and self-interaction for the hexagonal phase, respectively. More phonon dressing results in the destructive interaction, while less phonon dressing results in the constructive interaction of the atomic coherences. The experimental measurements of the photon–phonon interaction agree with the theoretical simulations. Based on our results, we proposed a model for an optical transistor (as an amplifier and switch).

## 1. Introduction

In the past years, it was desirable to couple a single atomic-like spin to a superconducting qubit, where a nanomechanical resonator is coupled to a two-level system to induce strong phonon–phonon interactions [1,2]. However, the entanglement generated is affected by different systems in a traditional method that often needs a strong spin–phonon interaction to exceed the decay of the phonons [3,4]. Phonon dispersion relation and lattice-spin coupling of Eu^3+^ have been reported [5,6]. A thermal phonon at elevated temperatures, lattice vibration structural transition, and thermal expansion behavior in LaPO_4_: Eu have also been studied [7].

Recently, the photon–phonon dressing coupling in Eu^3+^ ions doped BiPO_4_ has been studied [8,9], as Eu^3+^/Pr^3+^ ions are very sensitive to the site symmetry and its surrounding crystal field of the host material compared to other crystal ions [10,11,12]. Therefore, it can be achievable to obtain such a potential application in BiPO_4_ crystal. The crystal structure of BiPO_4_ has two polymorphic forms, monoclinic (M) and hexagonal (H) phases. The difference in the symmetry of the lattice structure results in different interactions [13,14]. The H phase of crystal is more structurally asymmetric than the M phase in Eu^3+^ because of a more atomic-like system. Bismuth phosphate (BiPO_4_) has drawn significant attention as a host medium for doping lanthanide ions due to its comparable ionic radius of Bi^3+^(1.11 Å) with that of lanthanide ions [15,16,17].

The Eu^3+^:BiPO_4_ is one of the most promising atomic-like mediums known for its long coherence time (ms) [8] due to photon–phonon coupling in doubly dressed states with potential applications in quantum memory [18,19,20].

Interactions of doubly dressed states and the corresponding properties of atomic systems have attracted considerable attention in recent decades. In this regard, two kinds of doubly dressed processes (in cascade- and nested-parallel schemes) were reported in an open five-level atomic system [21,22]. Nie et al. theoretically investigated the similarities and differences between different kinds of single dressing schemes for six-wave mixing to examine the interaction between multi-wave mixing in a five-level atomic system [23].

Next, we will consider such multi-nonlinear signals’ interaction with the coupling of a lattice vibration phonon and photon dressing. 

In this paper, we investigated two multi-dressing cross-interactions obtained from the various phase transitions of Eu^3+^:BiPO_4_ crystal by changing the time gates. The spectral cross-interaction evolves from out-of-phase FL, to hybrid (FL+SFWM), and to in-phase SFWM (anti-Stokes signal). Moreover, we demonstrate that the FL and SFWM destructive interaction results from more phonon dressing, and such dressing is achieved with a multiparameter temperature (300 K), H phase, and broadband excitation.

## 2. Experimental Scheme

The ion PO_4_^3−^: [Bi^3+^+Eu^3+^] has five molar ratios (7:1, 20:1, 6:1, 1:1, 0.5:1) for the Eu^3+^:BiPO_4_ sample with different lattice vibration structures. In our experiment (Figure 1a), we used five BiPO_4_ samples with different combinations and concentrations of a pure H phase and low-temperature monoclinic phase (LTMP), where the H phase refers to C_2_, and LTMP refers to the C_1_ site symmetry, respectively. The sample (7:1) corresponds to the pure M phase, (20:1) corresponds to the mixed [more M (75%) + less H (25%)] phase, (6:1) corresponds to the mixed [half H (50%) + half M (50%)] phase, (1:1) corresponds to the mixed [less H (25%) + more M (75%)] phase, and (0.5:1) corresponds to the pure H phase. The concentration of the Eu^3+^ ions is 5% consistent across all five samples with different phase transitions. Figure 1c shows the fine structure energy levels of the Eu^3+^:BiPO_4_ crystals. The Eu^3+^:BiPO_4_ has the ground state ^7^F_1_ and excited state ^5^D_0_ (mj = 0). The ground state ^7^F_1_ can split into mj = −1 (587.3 nm), mj = 0 (592.3 nm), and mj = +1 (597.3 nm) under the crystal field effect of the BiPO_4_ crystal and dressing effect.

To implement the experiment, Eu3+:BiPO4 samples were held in a cryostat (CFM-102). The temperature was controlled through liquid nitrogen from 300 K (large phonon Rabi frequency $G_{pi}^{T}$ with more thermal phonons) to 77 K, where $G_{pi}^{T}=-\mu_{kl}E_{pi}/\hbar$ is the Rabi frequency of the phonon field (I = 1, 2; T= (T1, T2) = (300 K, 77 K)). The $\mu_{kl}$ is the dipole moment between |*k*〉 and |*l*〉 of the crystal field splitting in the ^7^F_1_ state (Figure 1b), $E_{pi}$ is the phonon field, where such phonon builds atomic coherence for the crystal field splitting in ^7^F_1_.

In the experiment, the $G_{pi}^{T}$ and phase transition detuning $\Delta_{pi}^{j}$ are controlled by the temperature and different samples, respectively. The frequency detuning of the phonon field is $\Delta_{pi}^{j}=\Omega_{kl}-\omega_{pi}^{j}$(j = a (7:1), b (20:1), c (6:1), d (1:1), and e (0.5:1) sample), as shown in Figure 1b, where $\Omega_{kl}$ is the frequency between |*k*〉 and |*l*〉. The $\omega_{pi}^{j}$ is the phonon frequency of the phonon field, which is determined by the vibration frequency of the crystal lattice state mode. The different frequencies of the phase transitions ($\omega_{pi}^{a}<\omega_{pi}^{e}$, $\Delta_{pi}^{a}>\Delta_{pi}^{e}$) can couple to the different lattice vibrations for Eu^3+^:BiPO_4_, resulting in different phonon dressing ($|G_{pi}^{T}{|}^{2}/i\Delta_{pi}^{a}<|G_{pi}^{T}{|}^{2}/i\Delta_{pi}^{e}$).

Figure 1a shows the schematic diagram of the experimental setup. Here, we used two tunable dye lasers (narrow scan with a 0.04 cm^−1^ linewidth) pumped by an injection-locked single-mode Nd^3+^: YAG laser (Continuum Powerlite DLS 9010, 10 Hz repetition rate, 5 ns pulse width) to generate the pumping fields, broadband ***E*_1_** (*ω*_1_, Δ_1_) and narrowband ***E*_2_** (*ω*_2_, Δ_2_). The broadband excitation ***E*_1_** couples to more crystal field splitting levels ^5^D_0_ and ^7^F_1_ (Figure 1b), resulting in more lattice vibration (phonon dressing). However, the narrowband excitation ***E*_2_** couples to fewer splitting levels, resulting in less lattice vibration. The frequency detuning here is $\Delta_{i}=\Omega_{mn}-\omega_{i}$, where $\Omega_{mn}$ is the frequency between the crystal field splitting levels ^5^D_0_ and ^7^F_1_, and $\omega_{i}$ is the optical frequency. The Rabi frequency of the optical field is defined as $G_{i}=-\mu_{mn}E_{i}/\hbar$, where $\mu_{mn}$ is the dipole moment of the crystal field splitting with the different states ^5^D_0_ and ^7^F_1_ excited by the ***E*_i_** between the levels |*m*〉 and |*n*〉, as shown in Figure 1b. Such a photon builds the atomic coherence of the crystal field splitting with the different states (^5^D_0_ and ^7^F_1_). The pulse generated from the Nd^3+^: YAG laser is used to simultaneously trigger a boxcar-gated integrator and oscilloscope. The input laser beams are along the [010] axis of the BiPO_4_ crystal, which is perpendicular to the optical axis. The spectral optical outputs are obtained by scanning the laser frequency. The grating motor of the two dye lasers is scanned by a computer to form the x-axis, and the intensity of the excitation spectrum is the average of ten shots from the gated integrator (Figure 1a) appearing on the y-axis.

The optical signal generated from the Eu^3+^:BiPO_4_ crystal is detected via confocal lenses and photomultiplier tubes (PMTs). In our experimental setup, PMT1 is precisely placed to detect the narrowband FL and spontaneous four-wave mixing (SFWM) signal, whereas PMT2 is placed to detect the broadband FL and SFWM signal. Such a detector placement is based on the different distances from the detector to the sample (Figure 1a). Hence, the PMT affects the ratio of out-of-phase FL and in-phase SFWM. The out-of-phase FL1 signal and FL2 signals are generated through the excitation of the $E_{1}$ and $E_{2}$ lasers, respectively. The in-phase $E_{s1}$ signal is generated by a combination of the $E_{1}$ and reflection $E_{1}^{\prime }$ under the phase-matched condition ($k_{1}+k_{1}^{\prime }=k_{S}+k_{AS}$). At the same time, the spectral signals from the different energy levels with different lifetimes can be obtained through boxcar-gated integrators which can be controlled from the time gate. The time gate can control the ratio of out-of-phase FL and ***E*_S/AS_**.

Therefore, the photon–phonon atomic coherence interaction can be controlled by changing the time gate, broadband/narrowband excitation, and thermal/phase transition phonon.

### 2.1. Theoretical Model

#### 2.1.1. Photon–Photon Atomic Coherence Cross-Interaction 

The single laser or two lasers excitation shows photon dressing. Different lattice vibrations produced different frequency phonons. Such different frequency phonons can match to different crystal field splitting levels ^5^D_1_−^7^F_1_, ^5^D_0_−^7^F_1,_ and ^5^D_0_−^7^F_3_ in the ion Eu^3+^, so more phonon results in effective dressing. The three sharp dips are hard to be explained only by photon field dressing. Therefore, the phonon can be used to explain the three sharp dips. The cross-interactions, which evolve from FL to hybrid (coexistence of second order FL and SFWM), to SFWM are below
(1)$$\left|\rho_{F1}^{\left(2\right)}+\rho_{F2}^{\left(2\right)}{|}^{2}=|\rho_{F1}^{\left(2\right)}{|}^{2}+|\rho_{F2}^{\left(2\right)}{|}^{2}+2|\rho_{F1}^{\left(2\right)}||\rho_{F2}^{\left(2\right)}|\cos(\theta_{F}\right)$$
(2)$$\left|\rho_{AS1}^{\left(3\right)}+\rho_{AS2}^{\left(3\right)}{|}^{2}=|\rho_{AS1}^{\left(3\right)}{|}^{2}+|\rho_{AS2}^{\left(3\right)}{|}^{2}+2|\rho_{AS1}^{\left(3\right)}||\rho_{AS2}^{\left(3\right)}|\cos(\theta_{AS}\right)$$
(3)$$|\rho_{H}^{X}{|}^{2}=|\rho_{F1}^{\left(2\right)}+\rho_{F2}^{\left(2\right)}+\rho_{S1}^{\left(3\right)}+\rho_{S2}^{\left(3\right)}{|}^{2}$$

When the laser fields ***E*_1_** and ***E*_2_** are applied, the density matrix elements of out-of-phase FL for the [H+M]-phase Eu^3+^:BiPO_4_ via perturbation chain $\rho_{11}^{\left(0\right)}\overset{E_{1}}{\to }\rho_{12}^{\left(1\right)}\overset{{\left(E_{1}\right)}^{*}}{\to }\rho_{22}^{\left(2\right)}$ and $\rho_{00}^{\left(0\right)}\overset{E_{2}}{\to }\rho_{20}^{\left(1\right)}\overset{{\left(E_{2}\right)}^{*}}{\to }\rho_{22}^{\left(2\right)}$ can be written as $\rho_{F1}^{\left(2\right)}=-|G_{1}{|}^{2}/((\Gamma_{12}+i\Delta_{1}+|G_{2}{|}^{2}/(\Gamma_{02}+i(\Delta_{1}-\Delta_{2})))\Gamma_{22})$, $\rho_{F2}^{\left(2\right)}=-|G_{2}{|}^{2}/((\Gamma_{20}+i\Delta_{2}+|G_{1}{|}^{2}/(\Gamma_{21}-i(\Delta_{1}-\Delta_{2})))\Gamma_{22})$, where $\rho_{F1}^{\left(2\right)}=|\rho_{F1}^{\left(2\right)}|e^{i\theta_{1}}$, $\rho_{F2}^{\left(2\right)}=|\rho_{F2}^{\left(2\right)}|e^{i\theta_{2}}$, $\theta_{F}=\theta_{1}-\theta_{2}$. In the Λ-type three-level system, the third-order density matrix elements $\rho_{AS}^{\left(3\right)}$ via $\rho_{11}^{\left(0\right)}\overset{E_{1}}{\to }\rho_{21}^{\left(1\right)}\overset{E_{S}}{\to }\rho_{22}^{\left(2\right)}\overset{E_{1}^{\prime }}{\to }\rho_{20(AS)}^{\left(3\right)}$ can be written as $\rho_{AS}^{\left(3\right)}=-iG_{S}G_{1}G_{1}^{\prime }/((\Gamma_{21}+i\Delta_{1})(\Gamma_{22}+i\Delta_{1}+|G_{2}{|}^{2}/(\Gamma_{20}+i\Delta_{1}-i\Delta_{2}))(\Gamma_{20}+i\Delta_{1}+i\Delta_{1}^{\prime }))$, where $\rho_{AS1}^{\left(3\right)}=|\rho_{AS1}^{\left(3\right)}|e^{i\theta_{AS1}}$, $\rho_{AS2}^{\left(3\right)}=|\rho_{AS2}^{\left(3\right)}|e^{i\theta_{AS2}}$, $\theta_{AS}=\theta_{AS1}-\theta_{AS2}$. The $\Gamma_{ij}=(\Gamma_{i}+\Gamma_{j})/2$ is the transverse decay rate, where $\Gamma_{i/j}=\Gamma_{pop}+\Gamma_{ion-spin}+\Gamma_{ion-ion}+\Gamma_{phonon}+\Gamma_{dres\sin g}$. $\Gamma_{phonon}$ is more related to the broadband excitation.

In physics, the $\rho_{F1}^{\left(2\right)}$ generated from the field ***E*_1_** contains external field dressing $|G_{2}{|}^{2}$, and $\rho_{F2}^{\left(2\right)}$ from the field ***E*_2_** contains external field dressing $|G_{1}{|}^{2}$, as shown in Figure 2. Therefore, the $|\rho_{F1}^{\left(2\right)}+\rho_{F2}^{\left(2\right)}{|}^{2}$ shows the photon2 and photon1 dressing cross-interaction of the FL signal [16] at the profile ***E*_1_**/***E*_2_** resonance, as shown in Figure 3. In Equations (2) and (3), the $|\rho_{AS1}^{\left(3\right)}+\rho_{AS2}^{\left(3\right)}{|}^{2}$ and $\rho_{F1}^{\left(2\right)}+\rho_{F2}^{\left(2\right)}+\rho_{AS1}^{\left(3\right)}+\rho_{AS2}^{\left(3\right)}{|}^{2}$ are similar to the $|\rho_{F1}^{\left(2\right)}+\rho_{F2}^{\left(2\right)}{|}^{2}$ with two single external dressings. Therefore, the $|\rho_{AS1}^{\left(3\right)}+\rho_{AS2}^{\left(3\right)}{|}^{2}$ (Figure 4, Figure 5, Figure 6 and Figure 7) and $|\rho_{H}^{X}{|}^{2}$ (Figure 7) show the cross-interaction of the SFWM and hybrid signals, respectively. 

#### 2.1.2. Photon–Phonon Atomic Coherence Self-Interaction

The self-term $|\rho_{F2}^{\left(2\right)}{|}^{2}$ (or $|\rho_{F1}^{\left(2\right)}{|}^{2}$) is taken from in Equation (1) with the external dressing. The phonon1 dressing $|G_{p1}^{T}{|}^{2}$ and internal dressing $|G_{2}{|}^{2}$ (or $|G_{1}{|}^{2}$) are included in the self-term $|{\rho^{\prime \prime }}_{F2}^{\left(2\right)}{|}^{2}=|-|G_{2}{|}^{2}/((\Gamma_{20}+i\Delta_{2}+d_{1})\Gamma_{00}{|}^{2}$ with the broadband ***E*_1_** dressing (or $|{\rho^{\prime \prime }}_{F1}^{\left(2\right)}{|}^{2}=|-|G_{1}{|}^{2}/((\Gamma_{12}+i\Delta_{1}+d_{2})\Gamma_{22}{|}^{2}$ with the broadband ***E*_1_** generation), where $d_{1}=|G_{p1}^{T}{|}^{2}/(\Gamma_{10}+i\Delta_{2}-i\Delta_{p1}^{j})+|G_{1}{|}^{2}/(\Gamma_{21}+i\Delta_{2}-i\Delta_{1})$ (or $d_{2}=|G_{p1}^{T}{|}^{2}/(\Gamma_{10}+i\Delta_{1}+i\Delta_{p1}^{j})+|G_{2}{|}^{2}/(\Gamma_{02}+i\Delta_{1}-i\Delta_{2})$). For example, the $|{\rho^{\prime \prime }}_{F2}^{\left(2\right)}{|}^{2}$ with two cascade dressings is expanded as follows
(4)$$|{\rho^{\prime \prime }}_{F2}^{\left(2\right)}{|}^{2}=|\rho_{F2}^{\left(2\right)}+{\rho^{\prime }}_{F2}^{\left(4\right)}+{\rho^{\prime \prime }}_{F2}^{\left(4\right)}{|}^{2}$$

The $|{\rho^{\prime \prime }}_{F2}^{\left(2\right)}{|}^{2}$ and $|{\rho^{\prime \prime }}_{F1}^{\left(2\right)}{|}^{2}$ contain $\left|{\rho^{\prime \prime }}_{F2}^{\left(2\right)}{|}^{2}+2|{\rho^{\prime \prime }}_{F1}^{\left(2\right)}||{\rho^{\prime \prime }}_{F2}^{\left(2\right)}|\cos(\varphi_{F}^{\prime \prime }\right)$ in Equation (4) and $\left|{\rho^{\prime \prime }}_{F1}^{\left(2\right)}{|}^{2}+2|{\rho^{\prime \prime }}_{F1}^{\left(2\right)}||{\rho^{\prime \prime }}_{F2}^{\left(2\right)}|\cos(\varphi_{F}^{\prime \prime }\right)$, which show the out-of-phase FL2 and FL1 self-interaction of the two lasers, respectively. However, when the external field dressing is neglected at off-resonance, the Equation (4) becomes one laser self-interaction of FL.

The photon1 excites atomic coherence ($\Gamma_{12}$ and $\rho_{12}$) between |1〉 and |2〉 couples to the phonon1 atomic coherence by a common level |1〉 (Figure 1b) in $|{\rho^{\prime \prime }}_{F2}^{\left(2\right)}{|}^{2}$. The photon2 excites atomic coherence ($\Gamma_{20}$ and $\rho_{20}$) between |0〉 and |2〉. By using Taylor expansion for cascade dressing, the dressing (atomic coherence) coupling effect is transferred into the nonlinear generating process in Equation (4). Thus, we obtain the generating Hamiltonian $H=i\hbar \kappa_{F}\alpha_{1}^{†}\alpha_{2}^{†}\alpha_{p1}^{†}+H.c.$ for sixth-order nonlinearity, where $\kappa_{F}=-i\varpi_{F}\chi_{}^{\left(6\right)}E_{F}E_{1}^{}E_{2}^{}E_{p1}/2$. The $\varpi_{F}$ is the central frequency of FL.

Next, the difference from the self-term $|\rho_{AS2}^{\left(3\right)}{|}^{2}$ (or $|\rho_{AS1}^{\left(3\right)}{|}^{2}$) in Equation (2), the internal dressing $|G_{2}{|}^{2}$ (or $|G_{1}{|}^{2}$) and two phonon dressings ($|G_{p1}^{T}{|}^{2}$ and $|G_{p2}^{T}{|}^{2}$) are included in $\rho_{AS2}^{\prime \prime \prime \prime \left(3\right)}$ (or $\rho_{AS1}^{\prime \prime \prime \prime \left(3\right)}$). Where $\rho_{AS2}^{\prime \prime \prime \prime \left(3\right)}=-iG_{S2}G_{2}G_{2}/d_{0}$ (or $\rho_{AS1}^{\prime \prime \prime \prime \left(3\right)}=-iG_{S1}G_{1}{G^{\prime }}_{1}/((\Gamma_{21}+i\Delta_{1})d_{2}d_{3})$),

$d_{0}=(\Gamma_{20}+i\Delta_{2}+d_{1}+|G_{1}{|}^{2}/(\Gamma_{21}+i\Delta_{2}-i\Delta_{1}))(\Gamma_{22}+i\Delta_{2})(\Gamma_{20}+2i\Delta_{2})$, $d_{1}=|G_{2}{|}^{2}/(\Gamma_{20}+i\Delta_{1}+|G_{p1}^{T}{|}^{2}/(\Gamma_{01}+i\Delta_{1}-i\Delta_{p1}^{j}{+|G}_{p2}^{T}{|}^{2}/(\Gamma_{31}+i\Delta_{1}-i\Delta_{p1}^{j}+\Delta_{p2}^{j}))$, $d_{2}=\Gamma_{20}+i{\Delta^{\prime }}_{1}+i\Delta_{1}$, $d_{3}=\Gamma_{22}+i\Delta_{1}+d_{4}+d_{6}$, $d_{4}=|G_{2}{|}^{2}/(\Gamma_{20}+i\Delta_{1}+i\Delta_{2}+d_{5})$, $d_{5}=|G_{p1}^{T}{|}^{2}/(\Gamma_{01}+i\Delta_{1}+i\Delta_{2}-i\Delta_{p1}^{j}{+|G}_{p2}^{T}{|}^{2}/(\Gamma_{31}+i\Delta_{1}+i\Delta_{2}-i\Delta_{p1}^{j}+\Delta_{p2}^{j})$, $d_{6}=|G_{1}{|}^{2}/(\Gamma_{20}+2i\Delta_{1})$. The $|\rho_{AS1}^{\prime \prime \prime \prime \left(3\right)}{|}^{2}$ with the four cascade-nested dressing is expanded as follows
(5)$$|\rho_{AS1}^{\prime \prime \prime \prime \left(3\right)}{|}^{2}=|\rho_{AS1}^{\left(3\right)}+\rho_{AS1}^{\left(5\right)}+{\rho^{\prime }}_{AS1}^{\left(5\right)}+\rho_{AS1}^{\left(7\right)}+\rho_{AS1}^{\left(9\right)}{|}^{2}$$

The in-phase anti-Stokes $|\rho_{AS2}^{\prime \prime \prime \prime }{|}^{2}$ and $|\rho_{AS1}^{\prime \prime \prime \prime }{|}^{2}$ contain $\left|\rho_{AS2}^{\prime \prime \prime \prime \left(3\right)}{|}^{2}+2|\rho_{AS1}^{\prime \prime \prime \prime \left(3\right)}||\rho_{AS2}^{\prime \prime \prime \prime \left(3\right)}|\cos(\theta_{AS}^{\prime \prime \prime \prime }\right)$ in Equation (5) and $\left|\rho_{AS1}^{\prime \prime \prime \prime \left(3\right)}{|}^{2}+2|\rho_{AS1}^{\prime \prime \prime \prime \left(3\right)}||\rho_{AS2}^{\prime \prime \prime \prime \left(3\right)}|\cos(\theta_{AS}^{\prime \prime \prime \prime }\right)$, which show anti-Stokes2 and anti-Stokes1 self-interaction of the two lasers, respectively. When the external dressing is neglected at off-resonance, Equation (5) becomes one laser self-interaction of anti-Stokes.

The phonon1 excites atomic coherence ($\Gamma_{10}$ and $\rho_{10}$) between |0〉 and |1〉. The phonon2 excites atomic coherence ($\Gamma_{31}$ and $\rho_{31}$) between |1〉 and |3〉 (Figure 1b). In the four nested-cascade dressing of $\rho_{AS2}^{\prime \prime \prime \prime \left(3\right)}$, the atomic coherence from the nested coupling among the photon1, phonon1, and phonon2, couples with the atomic coherence of the photon2 (Figure 1b) in a cascaded manner. Similar to Equation (4), the dressing coupling effect is transferred into the nonlinear generating process in Equation (5). Thus, we also obtain the generating Hamiltonian which can be written as $H_{2}=i\hbar \kappa_{AS}\alpha_{1}^{†}\alpha_{2}^{†}\alpha_{p1}^{†}\alpha_{p2}^{†}+H.c.$ for ninth-order nonlinearity, where $\kappa_{S}=-i\varpi_{AS}{\chi^{\prime }}_{}^{\left(9\right)}E_{AS}E_{S}E_{1}^{}E_{2}^{}E_{p1}E_{p2}/2$. The $\varpi_{AS}$ is the central frequency of anti-Stokes.

#### 2.1.3. Simulation of Nonlinear Signals Dressing Interaction

Figure 2a shows the FL1 and FL2 self-terms $|\rho_{F1}^{\left(2\right)}{|}^{2}+|\rho_{F2}^{\left(2\right)}{|}^{2}$, the cross-term $2|\rho_{F1}^{\left(2\right)}||\rho_{F2}^{\left(2\right)}|\cos(\theta_{F})$ in the cross-interaction of the two lasers $|\rho_{sum}^{\left(2\right)}{|}^{2}$ at $\Delta_{1}=\Delta_{2}/2$ versus the detuning difference $\Delta =\Delta_{1}-\Delta_{2}$ from Equation (1), respectively. $|\rho_{F1}^{\left(2\right)}{|}^{2}$ and $|\rho_{F2}^{\left(2\right)}{|}^{2}$ have the maximal values at Δ=±4.1 THz and Δ=±3.6 THz, respectively. Hence, there exist two peaks at around Δ=±10 THz in the hot curve that represents the cross-interaction $|\rho_{sum}^{\left(2\right)}{|}^{2}$. The purple curve shows the cross-term $2|\rho_{F1}^{\left(2\right)}||\rho_{F2}^{\left(2\right)}|\cos(\theta_{F})$. Here, the value below or above zero suggests destructive or constructive interference, respectively. In fact, the variations of the phase difference between the second-order FL1 and FL2 change the constructive interaction into destructive interaction, and vice versa. Furthermore, by $\rho_{F1}^{\left(2\right)}=|\rho_{F1}^{\left(2\right)}|e^{i\theta_{1}}$ and $\rho_{F2}^{\left(2\right)}=|\rho_{F2}^{\left(2\right)}|e^{i\theta_{2}}$, we obtain $\left|\rho_{F1}^{\left(2\right)}+\rho_{F1}^{\left(2\right)}{|}^{2}-|\rho_{F1}^{\left(2\right)}{|}^{2}-|\rho_{F2}^{\left(2\right)}{|}^{2}=2|\rho_{F1}^{\left(2\right)}|\rho_{F1}^{\left(2\right)}|\cos(\theta_{F}\right)$ from Equation (1). Figure 2b shows the phases $\theta_{1}$, $\theta_{2}$, and the phase difference $\theta_{F}$ versus Δ as given in Table 1. As the $\theta_{1}$ and $\theta_{2}$ are changed, the $\theta_{F}$ alternates between −0.7π and 0.7π. The interaction switches from constructive ([−0.7π,0.5π)), destructive ([0.5π,0.7π)), constructive ([−0.5π,0.5π)), and destructive ([−0.7π,−0.5π)) and constructive ([−0.5π,0.7π)) as given in Table 2. Our simulation (Figure 2) is obtained by scanning $\Delta =\Delta_{2}-\Delta_{1}$ [23], and our experimental result (Figure 3, Figure 4, Figure 5, Figure 6 and Figure 7) is gained by scanning the dressing field $\Delta_{2}$. For simplicity, we only considered the external dressing in simulation Equation (1). Furthermore, Equations (1)–(3) reveal the cross-interaction of the two lasers. If the internal dressing and phonon dressing are considered, the cross-interaction becomes complicated.

### 2.2. Experiments

The photon excitation atomic coherence between the different states (^5^D_0_ and ^7^F_1_) can be coupled to the phonon excitation atomic coherence in the same state (^7^F_1_). Unlike the photon atomic coherence of the crystal field splitting with the different states, the phonon atomic coherence of the crystal field splitting in the same state is difficult to optically excite.

Moreover, the phonon dressing can control the destructive and constructive interaction. The constructive interaction results from less phonon dressing (77 K, M phase, narrowband ***E*_2_**), whereas the destructive interaction is caused by more phonon dressing (300 K, H phase, broadband ***E*_1_**).

#### 2.2.1. FL Dressing Cross- and Self-Interaction

Figure 3, Figure 4, Figure 5, Figure 6 and Figure 7 show the connected spectrum of the dressing cross-interaction of the two lasers with a different bandwidth. The spectrum profile of such interactions can be achieved by connecting several spectra together by scanning $\Delta_{2}/\Delta_{1}$ at a different detuning ($\Delta_{1}/\Delta_{2}$) and can be written as $\left|\rho_{F/AS1}^{}+\rho_{F/AS2}^{}{|}^{2}=R_{1}(\theta_{F/AS})+N_{1}(\theta_{F/AS})=R_{2}(\theta_{F/AS})+N_{2}(\theta_{F/AS}\right)$. When the $\Delta_{i}$ (*i* = 1, 2) is scanned, the $R_{i}$ and $N_{i}$ show a resonance and non-resonance profile term, respectively. The broad peak ($N_{i}(\theta_{F/AS}=0)$ profile) and broad dip ($N_{i}(\theta_{F/AS}=\pi )$ profile) in Figure 3, Figure 4, Figure 5, Figure 6 and Figure 7 show the constructive and destructive interaction, respectively.

Figure 3a,b,e,f show the constructive cross-interaction of FL (sharp peak $R_{i}(\theta_{F}=0)$, broad peak $N_{i}(\theta_{F}=0)$ (profile)) at the ***E*_1_/*E*_2_** resonance. When the time gate is fixed at 1 μs, the FL emission turns out to be dominant. The increasing sharp peaks at the ***E*_1_**
$N_{2}(\theta_{F}=0)$, as shown in Figure 3(a3,e3), and ***E*_2_**
$N_{1}(\theta_{F}=0)$, as shown in Figure 3(b3,f3), in off-resonance come from a constructive cross-interaction due to $|\rho_{F1}^{\left(2\right)}+\rho_{F2}^{\left(2\right)}{|}^{2}$ from Equation (1). Such an increasing sharp peak comes from the (6:1) sample and is recorded at a far detector position. Moreover, the broad peaks $N_{i}(\theta_{F}=0)$, as shown in Figure 3a,e,b,f, come from a single dressing constructive cross-interaction $N_{1}(\theta_{F}=0)$ and $N_{2}(\theta_{F}=0)$, respectively, which agrees with the two single external dressing simulations illustrated in Figure 2(a3). The sharp peaks at ***E*_1,_** as shown in Figure 3a,e and at ***E*_2,_** as shown in Figure 3b,f in off-resonance result from the self-interaction with the internal dressings $|G_{2}{|}^{2}$ and $|G_{1}{|}^{2}$, respectively.

Figure 3c shows the cross-interaction of FL (sharp dip $R_{2}(\theta_{F}^{\prime \prime }=\pi )$) and broad peak $N_{2}(\theta_{F}^{\prime \prime }=0)$) at the ***E*_1_** resonance. Compared with the sharp peak (Figure 3c) at the ***E*_1_** off-resonance, the dressing small dip $R_{2}(\theta_{F}^{\prime \prime }=\pi )$ at the ***E*_1_** resonance, Figure 3(c3), results from the switch of the two cascade dressings (external photon $|G_{1}{|}^{2}$ and phonon1 $\left|G_{p1}^{T1}{|}^{2}/(\Gamma_{10}+i\Delta_{p1}^{c})+|G_{1}{|}^{2}/(\Gamma_{20}+i\Delta_{1}\right)$ in Equation (4). Moreover, the sharp peak $R_{2}(\theta_{F}=0)$ at the ***E*_1_** resonance, Figure 3(a3), is transferred into a small dip $R_{2}(\theta_{F}^{\prime \prime }=\pi )$, as shown in Figure 3(c3), due to phonon1 dressing $\left|G_{p1}^{T1}{|}^{2}/(\Gamma_{10}+i\Delta_{p1}^{c}\right)$ at the near detector position (broadband FL). Similar to Figure 3a,b,e,f, the broad peak, as shown in Figure 3c, results from the constructive cross-interaction $N_{2}(\theta_{F}^{\prime \prime }=0)$ with less phonon dressing.

Figure 3d,g,h show the destructive cross-interaction of FL (sharp dressing dip $R_{i}(\theta_{F}^{\prime \prime }=\pi )$), broad dip $N_{i}(\theta_{F}^{\prime \prime }=\pi )$ (profile)) at the ***E*_1_** resonance. Compared with the sharp dip at the ***E*_2_** off-resonance from the destructive self-interaction (Figure 3d), the sharp dip at the ***E*_2_** resonance, as shown in Figure 3(d3), increases due to the destructive cross-interaction $R_{1}(\theta_{F}^{\prime \prime }=\pi )$ with the external dressing $|G_{2}{|}^{2}$ of ${\rho^{\prime \prime }}_{F1}^{\left(2\right)}$ in $|{\rho^{\prime \prime }}_{F1}^{\left(2\right)}+\rho_{F2}^{\left(2\right)}{|}^{2}$ at the broadband excitation ***E*_1_** and 300 K. This is because the more crystal field splitting levels ^7^F_1_ (Figure 1b) and lattice vibrations are coupled by broadband excitation ***E*_1_**. Moreover, the 300 K results in more thermal phonons with large $G_{p1}^{T1}$. The broad dip (Figure 3d) comes from a stronger destructive cross-interaction $N_{1}(\theta_{F}^{\prime \prime }=\pi )$ with more phonon dressing. The sharp dressing dips at the ***E*_2_** off-resonance (Figure 3d) come from the self-interaction from Equation (4).

The sharp dips at the ***E*_1_** off-resonance come from the phonon1 dressing $\left|G_{p1}^{T1}{|}^{2}/(\Gamma_{10}+i\Delta_{p1}^{c}\right)$ of ${\rho^{\prime \prime }}_{F1}^{\left(2\right)}$, as shown in Figure 3d. The sharp dip $R_{1}(\theta_{F}^{\prime \prime }=\pi )$ at the ***E*_1_** resonance results from the cascade dressing $\left|G_{2}{|}^{2}+|G_{p1}{|}^{2}/(\Gamma_{10}+i\Delta_{2}+i\Delta_{p1}^{d}\right)$ of ${\rho^{\prime \prime }}_{F1}^{\left(2\right)}$, as shown in Figure 3(d3). Such a cascade dressing coupling results in photon1–phonon2–phonon1 ($\alpha_{1}^{†}$, $\alpha_{2}^{†}$, $\alpha_{p1}^{†}$ in $\chi^{\left(6\right)}$) atomic coherence coupling.

Figure 3g corresponds to the simulation (Figure 5g) modelled through Equation (4). Compared with the sharp dip at the ***E*_1_** off-resonance, Figure 3g, the sharp dip at the ***E*_1_** resonance, Figure 3(g3), decreases due to the cross-interaction $R_{2}(\theta_{F}^{\prime \prime }=\pi )$ with the phonon1 dressing $|G_{p1}^{T1}{|}^{2}/(\Gamma_{10}+i\Delta_{p1}^{d})+|G_{1}{|}^{2}$ from Equation (4) at the narrowband excitation. The broad dip, as shown in Figure 3g, comes from the strong destructive cross-interaction $N_{2}(\theta_{F}^{\prime \prime }=\pi )$ with more phonon dressing. Similar to Figure 3d, the broad dip (Figure 3h) can be explained by a stronger destructive cross-interaction $N_{1}(\theta_{F}^{\prime \prime }=\pi )$ with the phonon1 dressing $|G_{p1}^{T1}{|}^{2}/(\Gamma_{10}+i\Delta_{p1}^{d})+|G_{2}{|}^{2}$ of ${\rho^{\prime \prime }}_{F1}^{\left(2\right)}$. Compared with the small dip at the ***E*_1_** resonance, Figure 3(c3), the large dip is, as shown in Figure 3(g3), due to the phase transition phonon dressing $\left|G_{p1}^{T1}{|}^{2}/(i\Delta_{p1}^{c}+i\Delta_{1})<|G_{p1}^{T1}{|}^{2}/(i\Delta_{p1}^{d}+i\Delta_{1}\right)$. Such a phonon dressing dip results from the resonance detuning $\Delta_{p1}^{d}$($\Delta_{p1}^{d}<\Delta_{p1}^{c}$), which is due to the high phonon frequencies $\omega_{p1}^{d}$ ($\omega_{p1}^{d}>\omega_{p1}^{c}$) for the H-phase samples (6:1, 1:1), as shown in Figure 3d,h.

The experimental setup presented in Figure 1a is used to realize the optical transistor as an amplifier and switch (Figure 1d) where the Eu^3+^:BiPO_4_ crystal behaves as a transistor with the ***E*_1_** beam as its input (a_in_); the ***E*_2_** is a control signal, a_out_ is the output of the transistor detected at PMT. The transistor gain (g) depends upon the external dressing effect which can be controlled through the detuning of the ***E*_2_** beam [24,25]. In Figure 1d, the transistor as a peak amplifier1 and dip amplifier2 are realized from the spectral intensity results presented in Figure 3b,d, respectively. The signal amplification (peak or dip) results from photon dressing and varies with experimental parameters such as the position of the PMT, laser detuning (bandwidth) and Eu^3+^:BiPO_4_ sample. At a far PMT (Figure 3b), only a peak amplication (amplifier1) is observed as FL is weak (no dressing) whereas strong FL (strong dressing) at a near PMT shows a dip amplication (amplifier2), as shown in Figure 3d. By exploring the relationship between the transistor amplifier and laser bandwidth, we observed that the narrowband laser ***E*_2_** (Figure 3a) has a higher transistor gain than the broadband laser ***E*_1_** (Figure 3b). Furthermore, our results show that (6:1) the Eu^3+^:BiPO_4_ sample has a higher amplication factor than (1:1) Eu^3+^:BiPO_4_.

In contrast to the amplifer, the transistor switch results from the photon–phonon atomic coherence interaction strongly depend upon several exprimental parameters such as the sample temperature, laser detuning (bandwidth, power), and molar ratio of the Eu^3+^:BiPO_4_ sample. For example, the (0.5:1) BiPO_4_ sample has more phonons due to strong lattice vibrations compared to the (0.5:1) BiPO_4_ sample. In addition, a higher temperature will result in more phonons, resulting in prominent spectral switching. To understand the workings of a transistor as an amplifier, we set the ***E*_2_** at off-resonance ($\Delta_{2}\neq 0$), then the amplitude of both the sharp peak in Figure 3(b1), and sharp dip in Figure 3(d1), are very low. When the detuning of the ***E*_2_** approaches resonance ($\Delta_{2}=0$), the sharp peak in Figure 3(b3), and dip in Figure 3(d3), amplifies by a factor. The amplification of the spectral signals can be explained by the high gain (g = 3.6) caused by the strong external dressing $|G_{2}{|}^{2}$ at the resonance wavelength.

Next, we extend our research and study the cross-interaction of SFWM in the following Section 2.2.2.

#### 2.2.2. SFWM Dressing Cross- and Self-Interaction

The out-of-phase FL (time gate = 1 μs) interaction is transferred to the in-phase SFWM interaction (time gate = 500 μs). When the time gate is increased to 5 μs, the SFWM signal (sharp $R_{i}$) is dominant.

Figure 4a shows the constructive cross-interaction of SFWM (two sharp peaks $R_{1}(\theta_{AS}=0)$, broad peak $N_{1}(\theta_{AS}=0)$ (profile)) at the ***E*_1_** resonance. Compared with the two sharp peaks at the ***E*_2_** off-resonance (Figure 4a), the two sharp peaks, Figure 4(a3), at the ***E*_2_** resonance also increase due to the constructive interaction $R_{1}(\theta_{AS}=0)$ with $|\rho_{AS1}^{\left(3\right)}+\rho_{AS2}^{\left(3\right)}{|}^{2}$ in Equation (2). Such two sharp peaks can be explained by the crystal field splitting levels (|1〉,|0〉) due to the high resolution of in-phase SFWM. The broad peak comes from the cross-interaction $N_{1}(\theta_{AS}=0)$, as shown in Figure 4a.

Figure 4b,d show the constructive cross-interaction of SFWM (single sharpest peak $R_{2}(\theta_{AS}^{\prime }=0)$, broad peak $N_{2}(\theta_{AS}^{\prime }=0)$) at the ***E*_1_** resonance. Compared with the sharpest peak at the ***E*_1_** off-resonance (Figure 4b), the amplitude of the sharpest peak $R_{2}(\theta_{AS}^{\prime }=0)$ at the ***E*_1_** resonance, Figure 4(b3), decreases due to the constructive cross-interaction with the phonon1-assisted $G_{2}$ dressing ($G_{p1}^{T2}$ and $G_{2}$ share the common atomic coherence) ${\left(|G_{1}{|}^{2}+|G_{p1}^{T}|\right)}^{2}/(\Gamma_{10}+i\Delta_{1}+i\Delta_{p1}^{a})$ of ${\rho^{\prime }}_{AS2}^{\left(3\right)}$. Compared with the sharp peak, Figure 4(a3), the linewidth of such a sharp peak decreases due to less thermal phonons (77 K) with a small $G_{p1}^{T2}$. The broad peak comes from the constructive interaction $N_{2}(\theta_{AS}^{\prime }=0)$, as shown in Figure 4b, with less phonon dressing. However, the sharpest peak, Figure 4b, at the ***E*****_1_** off-resonance is due to the self-interaction.

Figure 4c shows the constructive cross-interaction of SFWM (two sharp peaks $R_{1}(\theta_{AS}^{\prime \prime }=0)$, broad peak $N_{1}(\theta_{AS}^{\prime \prime }=0)$) at the ***E*_2_** resonance. The proportion of the two sharp peaks accounts for roughly 80% and the proportion of the single sharp dips only accounts for roughly 20%, as shown in Figure 4c. Compared with the two sharp peaks (the left peak from splitting energy level mj=−1; the right peak from splitting energy level mj=0) at the ***E*_2_** off-resonance, as shown in Figure 4c, two such sharp peaks, Figure 4(c3), at the ***E*_2_** resonance decrease due to the constructive cross-interaction $R_{1}(\theta_{AS}^{\prime \prime }=0)$. The broad peak (Figure 4c) comes from the constructive interaction $N_{1}(\theta_{AS}^{\prime \prime }=0)$. The small sharp dip, Figure 4(c3), at the ***E*_2_** resonance is obtained from the destructive cross-interaction due to the phonon1 dressing $|G_{p1}^{T}{|}^{2}/(\Gamma_{10}+i\Delta_{2}+i\Delta_{p1}^{b})+|G_{2}{|}^{2}$ in Equation (5).

Compared with the two sharp peaks at the ***E*_2_** resonance for the (7:1) sample, Figure 4(a3), the small sharp dip at the ***E*_2_** resonance, as shown in Figure 4(c3), results from the more phonon dressing ($\Delta_{p1}^{a}>\Delta_{p1}^{b}$, $|G_{p1}^{T2}{|}^{2}/i\Delta_{p1}^{a}<|G_{p1}^{T2}{|}^{2}/i\Delta_{p1}^{b}$) for H-sample (20:1). Such a small sharp dip results from the switch of the two cascade dressings (external photon $|G_{2}{|}^{2}$ and phonon1 $|G_{p1}^{T1}{|}^{2}/(\Gamma_{10}+i\Delta_{p1}^{b})+|G_{2}{|}^{2}$ of ${\rho^{\prime }}_{AS1}^{\left(3\right)}$), because the phonon dressing is easily distinguished by in-phase SFWM.

Similar to Figure 4(b3), the sharpest peak at the ***E*_1_** resonance also decreases due to the constructive cross-interaction $R_{2}(\theta_{AS}^{\prime }=0)$ with the phonon1-assisted dressing, as shown in Figure 4(d3).

Figure 5a shows the constructive cross-interaction of SFWM (sharp peak $R_{2}(\theta_{AS}=0)$), and the broad peak $N_{2}(\theta_{AS}=0)$) at the ***E*_1_** resonance. Similar to Figure 4b,d, the sharp peak, Figure 5(a3), at the ***E*_1_** resonance decreases compared to the sharp peaks at the ***E*_1_** off-resonance, Figure 5a, due to the cross-interaction $R_{2}(\theta_{AS}=0)$ with the phonon1-assisted dressing ${\left(|G_{1}{|}^{2}+|G_{p1}^{T1}|\right)}^{2}/(\Gamma_{10}+i\Delta_{1}+i\Delta_{p1}^{c})$ of ${\rho^{\prime }}_{AS2}^{\left(3\right)}$. The broad peak at the ***E*_1_** off-resonance comes from the constructive interaction $R_{2}(\theta_{AS}=0)$ due to less phonon dressing.

Figure 5b shows the constructive cross-interaction of SFWM (two sharp peaks $R_{1}(\theta_{AS}^{\prime \prime }=0)$, broad peak $N_{1}(\theta_{AS}^{\prime \prime }=0)$) at the ***E*_2_** resonance. Compared with the small dip at the ***E*_2_** resonance, Figure 4(c3), the small dip at the ***E*_2_** resonance, Figure 5(b3), increases due to the cross-interaction $R_{1}(\theta_{AS}^{\prime \prime }=0)$ with the phonon1 dressing $\left|G_{p1}^{T1}{|}^{2}/(\Gamma_{10}+i\Delta_{2}+i\Delta_{p1}^{c}\right)$ in Equation (5). Such a small sharp dip results from the switch of the two cascade dressings $|G_{p1}^{T1}{|}^{2}/(\Gamma_{10}+i\Delta_{p1}^{c})+|G_{2}{|}^{2}$ of ${\rho^{\prime \prime }}_{AS1}^{\left(3\right)}$. The broad peak, as shown in Figure 5b, comes from the constructive cross-interaction $N_{1}(\theta_{AS}^{\prime \prime }=0)$ due to less phonon dressing.

Figure 5c shows the cross-interaction of SFWM (sharp peak $R_{2}(\theta_{AS}^{\prime \prime \prime }=0)$, broad dip $N_{2}(\theta_{AS}^{\prime \prime \prime }=0)$ (profile)) at the ***E*_1_** resonance. The sharp dip at the ***E*_1_** off-resonance (Figure 5c) is transferred into the sharp peak at the ***E*_1_** resonance, Figure 5(c3), due to the constructive cross-interaction $R_{2}(\theta_{AS}^{\prime \prime \prime }=0)$ with the phonon1 dressing $|G_{p1}^{T1}{|}^{2}+|G_{1}{|}^{2}$ of ${\rho^{\prime \prime }}_{AS1}^{\left(3\right)}$ at the narrowband excitation. Such a transition (sharp dip $R_{2}(\theta_{AS}^{\prime \prime \prime }=\pi )$ to a sharp peak $R_{2}(\theta_{AS}^{\prime \prime \prime }=0)$) results from the switch of the three cascade dressings (internal photon $G_{2}$, external photon $G_{1}$ and phonon1 $G_{p1}$ of $|G_{p1}^{T1}{|}^{2}/(\Gamma_{10}+i\Delta_{p1}^{c})+|G_{1}{|}^{2}+|G_{2}{|}^{2}$ in $|\rho_{AS1}^{\left(3\right)}+{\rho^{\prime \prime \prime }}_{AS2}^{\left(3\right)}{|}^{2}$). The broad dip comes from the strong constructive interaction $N_{1}(\theta_{AS}^{\prime \prime \prime }=\pi )$, as shown in Figure 5c, with more phonon dressing. More interestingly, the sharp dips at the ***E*_1_** off-resonance (Figure 5c) result from the self-interaction in Equation (5). Such sharp dips are obtained from 300 K due to more thermal phonon dressing (large $G_{p1}^{T1}$). Compared with the sharpest peak at the ***E*_1_** off-resonance for the (7:1) and (20:1) samples (Figure 4c,d), the sharp dip at the ***E*_1_** off-resonance (Figure 5c) decreases due to the phase transition phonon dressing ($|G_{p1}^{T1}{|}^{2}/i\Delta_{p1}^{c}>|G_{p1}^{T2}{|}^{2}/i\Delta_{p1}^{a,b}$) for (6:1) more H-phase sample.

Figure 5d shows the destructive cross-interaction of SFWM (three sharp dips $R_{1}(\theta_{AS}^{\prime \prime \prime \prime }=\pi )$, broad dip $N_{1}(\theta_{AS}^{\prime \prime \prime \prime }=\pi )$) at the ***E*_2_** resonance. The differences with the three sharp dips at the ***E*_2_** off-resonance from the destructive self-interaction in Equation (5), as shown in Figure 5d, and the three sharp dips at the ***E*_2_** resonance, as shown in Figure 5(d3), result from the destructive cross-interaction $R_{1}(\theta_{AS}^{\prime \prime \prime \prime }=\pi )$ with the phonon1 and phonon2 dressing. The broad dip, as shown in Figure 5d, is obtained from the stronger destructive cross-interaction $N_{1}(\varphi_{AS}^{\prime \prime \prime \prime }=\pi )$ with more phonon dressing at 300 K and the broadband excitation. Figure 5d corresponds to the simulation result (Figure 5h) from Equation (5).

The three sharp dips at the ***E*_2_** off-resonance (Figure 5d) result from the three nested dressings (internal photon $G_{1}$, two phonons). However, the decreasing three sharp dips $R_{1}(\theta_{AS}^{\prime \prime \prime \prime }=\pi )$ at the ***E*_2_** resonance, Figure 5(d3), come from the external dressing $|G_{2}{|}^{2}$ of the four nested-cascade dressings (internal photon $G_{1}$, external photon $G_{2}$ and two phonons) $|G_{1}{|}^{2}/(\Gamma_{20}+|G_{p1}^{T1}{|}^{2}/(\Gamma_{10}-i\Delta_{p1}^{c}{+|G}_{p2}^{T1}{|}^{2}/(\Gamma_{13}-i\Delta_{p1}^{c}-i\Delta_{p2}^{c})))+|G_{2}{|}^{2}$ of $\rho_{AS1}^{\prime \prime \prime \prime \left(3\right)}$ Equation (5) in $|\rho_{AS2}^{\left(3\right)}+\rho_{AS1}^{\prime \prime \prime \prime \left(3\right)}{|}^{2}$. Such four dressing coupling results from the photon1–phonon2–phonon1–phonon2 ($\alpha_{1}^{†},\alpha_{2}^{†},\alpha_{p1}^{†},\alpha_{p2}^{†}$ in $\chi^{\left(9\right)}$) atomic coherence coupling.

More phonon dressing results from more lattice vibrations at 300 K for Eu^3+^:BiPO_4_ than the other samples (Eu^3+^/Pr^3+^: YPO_4_ [24] and Pr^3+^: Y_2_SO_5_ [26]). The model for the phonon-controlled transistor switch is presented, as shown in Figure 1d, where ‘enhancement peak’ and ‘suppression dip’ correspond to ‘ON-state’ and ‘OFF-state’, respectively. When the input signal (Figure 5b) is at a single ON-State (higher than baseline), then the corresponding output signal (Figure 5d) is at a single OFF-State (lower than baseline). Such a spectral switch can be controlled by single phonon dressing ($|G_{p1}{|}^{2}$). Our experimental results defined the switching contrast as *C=* |*I_off_ –I_on_*|*/ (I_off_ +I_on_)*, where *I_off_* is the intensity at the OFF-state and *I_on_* is the intensity at the ON-state. The maximum switching contrast *C* for a single state switch is about 82%, as shown in Figure 3(b3,d3). Furthermore, when the ON-state of the input signal is observed with two sharp peaks, Figure 5(b3), the corresponding output signal has the OFF-state with three sharp dips, as observed in Figure 5(d3). Such a multi-states switch can be controlled by two phonon dressings ($|G_{p1}{|}^{2}$, $|G_{p2}{|}^{2}$). The switching contrast *C* is about 93.6% for the multi-states switch measure for Figure 5(b3) and Figure 5(d3).

Figure 5e,f show the constructive cross-interaction of SFWM (single sharpest peak $R_{2}(\theta_{AS}^{\prime }=0)$, broad peak $N_{2}(\theta_{AS}^{\prime }=0)$) at the ***E*_1_** resonance. Compared with the sharpest peaks at the ***E*_1_** off-resonance, as shown in Figure 5e, the sharpest peak $R_{2}(\theta_{AS}^{\prime }=0)$ at the ***E*_1_** resonance, as shown in Figure 5(e3), increases due to the constructive cross-interaction with the phonon1-assisted dressing ${\left(|G_{1}{|}^{2}+|G_{p1}^{T2}|\right)}^{2}/(\Gamma_{10}+i\Delta_{1}+i\Delta_{p1}^{c})$ of ${\rho^{\prime }}_{AS2}^{\left(3\right)}$ at 77 K. Figure 5g shows the simulation results corresponding to the experimental results (Figure 5b). The transition from the broad dip $N_{2}(\theta_{AS}^{\prime }=\pi )$ (Figure 5c) to broad peak $N_{2}(\theta_{AS}^{\prime }=0)$ (Figure 5f) is due to the reduction of phonon dressing. Therefore, thermal phonon dressing plays a key role in the cross-interaction.

In order to explore further, we further compare the FL and SFWM interaction in Section 2.2.3.

#### 2.2.3. Comparison of FL and SFWM Interaction

The cross-interaction, as shown in Figure 6 and Figure 7, evolves from out-of-phase FL to hybrid (coexistence of second order FL and SFWM), to in-phase SFWM by changing the time gate (1 μs to 500 μs) obtained from the (0.5:1) sample.

Figure 6a,b show the constructive cross-interaction of FL (sharp peak $R_{1}(\theta_{F}=0)$, broad peak $N_{1}(\theta_{F}=0)$) at the ***E*_2_** resonance. Similar to Figure 3(a3,b3,e3,f3), the increasing sharp peaks, Figure 6(a3), at the ***E*_2_** resonance are due to the constructive cross-interaction $R_{1}(\theta_{F}=0)$ in Equation (1). Compared with the sharp peak at the ***E*_2_** off-resonance (Figure 6b), the sharp peak $R_{1}(\theta_{F}=0)$ at the ***E*_2_** resonance, Figure 6(b3), decreases due to the phonon1-assisted dressing ${\left(|G_{2}{|}^{2}+|G_{p1}^{T1}|\right)}^{2}/(\Gamma_{10}+i\Delta_{2}+i\Delta_{p1}^{e})$ of ${\rho^{\prime }}_{F1}^{\left(3\right)}$ (similar to Figure 5a).

Figure 6c,d show the constructive cross-interaction of SFWM (two sharpest peaks $R_{1}(\theta_{AS}^{\prime }=0)$, broad peak $N_{1}(\theta_{AS}^{\prime }=0)$) at the ***E*_2_** resonance. When the time gate increases to 500 μs, compared with the two sharpest peaks at the ***E*_2_** off-resonance, as shown in Figure 6c,d, the two sharpest peaks $R_{1}(\theta_{AS}^{\prime }=0)$ at the ***E*_2_** resonance, as shown in Figure 6(c3,d3), decrease due to the phonon1-assisted dressing ${\left(|G_{2}{|}^{2}+|G_{p1}^{T2}|\right)}^{2}/(\Gamma_{10}+i\Delta_{2}+i\Delta_{p1}^{e})$ in ${\rho^{\prime }}_{AS1}^{\left(3\right)}$(similar to Figure 4(c3) and Figure 5(a3)). The spectral linewidth of the sharp peak, as shown in Figure 6(a1), at 300 K is nine times larger than the linewidth at 77 K, as shown in Figure 6(c1), due to more thermal phonon dressing ($|G_{p1}^{T1}{|}^{2}/i\Delta_{p1}^{e}>|G_{p1}^{T2}{|}^{2}/i\Delta_{p1}^{e}$).

Similar to the sharp peak with the (6:1) sample at 300 K (large $G_{p1}^{T1}$), as shown in Figure 5a,b, the two sharpest peaks with the (0.5:1) sample are also shown at 77 K (small $G_{p1}^{T2}$), as shown in Figure 6c,d due to the phase phonon dressing $\left|G_{p1}^{T2}{|}^{2}/(i\Delta_{p1}^{e}+i\Delta_{1}\right)$ from the resonance detuning ($\Delta_{p1}^{e}\approx 0$). Therefore, compared with out-of-phase FL (Figure 6a,b), in-phase SFWM is more sensitive to phonon dressing (Figure 6c,d).

Figure 7a,b show the constructive cross-interaction of FL (sharp peak $R_{2}(\theta_{F}=0)$, broad peak $N_{2}(\theta_{F}=0)$) at the ***E*_1_** resonance. When the time gate is fixed at 1 μs, compared with the sharp peak at the ***E*_1_** off-resonance, as shown in Figure 7a, the sharp peak $R_{2}(\theta_{F}=0)$ at the ***E*_1_** resonance, as shown in Figure 7(a3), increases (similar to Figure 6a). The sharp peak $R_{2}(\theta_{F}=0)$ at the ***E*_1_** resonance, as shown in Figure 7(b3), decreases compared to the sharp peaks at the ***E*_1_** off-resonance, as shown in Figure 7b, due to the phonon1-assisted dressing ${\left(|G_{1}|+|G_{p1}^{T1}|\right)}^{2}/(\Gamma_{21}+i\Delta_{1}+i\Delta_{p1}^{e})$ of ${\rho^{\prime }}_{F2}^{\left(2\right)}$.

Figure 7c,d show the constructive cross-interaction of the hybrid (single sharpest peak $R_{2}$, broad peak $N_{2}$) at the ***E*_1_** resonance. When the time gate increases to 100 μs, compared to the sharpest peak at the ***E*_1_** off-resonance, as shown in Figure 7c,d, the sharpest peaks $R_{2}$ at the ***E*_1_** resonance, as shown in Figure 7(c3, d3), decrease due to the constructive cross-interaction with the phonon1-assisted dressing of $R_{2}$ in Equation (3). The broad peaks $N_{2}$, as shown in Figure 7c,d, can be explained by the constructive cross-interaction with less phonon dressing.

Figure 7e shows the destructive cross-interaction of SFWM (single sharpest dips $R_{2}(\theta_{AS}^{\prime \prime }=\pi )$, broad dip $N_{2}(\theta_{AS}^{\prime \prime }=\pi )$) at the ***E*_1_** resonance. When the time gate increases to 500 μs, compared to the difference with the sharpest dip at the ***E*_1_** off-resonance, the sharpest dip $R_{2}(\theta_{AS}^{\prime \prime }=\pi )$ at the ***E*_1_** resonance, as shown in Figure 7(e3), decreases due to the stronger destructive cross-interaction with the phonon1 dressing. Such a decrease in the sharpest dip $R_{2}(\theta_{AS}^{\prime \prime }=\pi )$ results from the external dressing $|G_{1}{|}^{2}$ of the two cascade dressings $|G_{p1}^{T1}{|}^{2}/(\Gamma_{10}+i\Delta_{p1}^{d})+|G_{1}{|}^{2}$ in ${\rho^{\prime \prime }}_{AS2}^{\left(3\right)}$. Moreover, the broad dip $N_{2}(\theta_{AS}^{\prime \prime }=\pi )$ is obtained from 300 K and the (0.5:1) sample with more phonon dressing.

Similar to Figure 5c, Figure 7f shows the cross-interaction of SFWM (single sharpest peak $R_{2}(\theta_{AS}^{\prime \prime }=0)$, broad dip $N_{2}(\theta_{AS}^{\prime \prime }=\pi )$ at the ***E*_1_** resonance. Compared to the sharp dip at the ***E*_1_** off-resonance, as shown in Figure 7f, the sharpest peak at the ***E*_1_** resonance, as shown in Figure 7(f3), comes from the constructive cross-interaction $R_{2}(\theta_{AS}^{\prime \prime }=0)$ with less phonon dressing. Such a transition (sharpest dip $R_{2}(\theta_{AS}^{\prime \prime }=\pi )$ to the sharpest peak $R_{2}(\theta_{AS}^{\prime \prime }=0)$) results from the switch of the three cascade dressings $|G_{p1}^{T1}{|}^{2}/(\Gamma_{10}+i\Delta_{p1}^{e})+|G_{1}{|}^{2}+|G_{2}{|}^{2}$ of ${\rho^{\prime \prime \prime }}_{AS2}^{\left(3\right)}$. Similar to Figure 5c, the broad dip in Figure 7f, comes from the strong destructive cross-interaction $N_{2}(\theta_{AS}^{\prime \prime }=\pi )$ with more phonon dressing.

Therefore, the out-of-phase FL constructive interaction (Figure 7a,b) can be evolved to the in-phase SFWM destructive interaction (Figure 7e,f). The H-phase result (Figure 5 and Figure 6) comes from sensitive phonon dressing and easy distinction for in-phase SFWM.

Moreover, the linewidth of the peak increases from 0.4±0.1 nm, as shown in Figure 7c, to 4.7±0.1 nm, as shown in Figure 7a, due to the $\Gamma_{phonon}$ of the generating process. The width of the dressing dip increases from 0.6±0.1 nm, as shown in Figure 5d, to 5.9±0.2 nm, as shown in Figure 3d, due to the $\Gamma_{phonon}$ of the dressing process. The destructive cross-interaction $R_{1}$, as shown in Figure 3d and d, results from more phonon dressing with the same area. However, such more phonon dressing shows different phenomena for the single sharp FL dip, as shown in Figure 3d, and the three sharpest SFWM dips, as shown in Figure 5d.

## 3. Discussion

From our results, we conclude that, unlike cross-interaction $N_{i}$ [non-resonance], the internal and external dressing atomic coherence coupling results in switching between constructive to destructive for $R_{i}$ (Figure 3c, Figure 4b, Figure 5b,c and Figure 7f). The resonant cross-interaction $R_{i}$ is distinguished from the non-resonant cross-interaction $N_{i}$ without internal dressing.

Furthermore, the destructive interactions result from cascade dressing (Figure 3d–f, Figure 4c, Figure 5b,c and Figure 7e) and four nested-cascade dressing (Figure 5d), respectively. The cascade dressing and nested dressing suggest strong and stronger photon–phonon atomic coherence coupling (leading to the three dressing dips, as shown in Figure 5d), respectively.

## 4. Conclusions

In summary, we theoretically and experimentally studied the constructive and destructive photon–phonon atomic coherence interaction. The destructive spectral interactions result from cascade dressing and four nested-cascade dressing, respectively. Here, we controlled the spectral interactions through the temperature, laser detuning/bandwidth, and molar ratio of the Eu^3+^:BiPO_4_ crystal. Due to more phonon dressing caused by high lattice vibrations, H-phase BiPO_4_ shows strong photon–phonon atomic coherence interaction. The cascade dressing with strong photon–phonon atomic coherence coupling led to the single sharp dip. Moreover, the four nested-cascade dressing with stronger photon–phonon atomic coherence coupling led to three sharp dips. Furthermore, the spectral evolution of the spectral cross-interaction from out-of-phase FL, to hybrid, and to in-phase SFWM is controlled by the time gates. The experimental results were verified through theoretical simulations. From our experimental results, the optical transistor (amplifier and switch) was also realized from the photon–phonon atomic coherence interaction where the signal amplification (peak or dip) is controlled by photon dressing and the switch results by the photon–phonon interaction. By controlling the phase transition, laser detuning, and temperature, a high amplifier gain of about 3.6 and switching contrast (93.6%) is achieved from our proposed technique.

## Figures and Tables

**Figure 1 nanomaterials-12-04304-f001:**
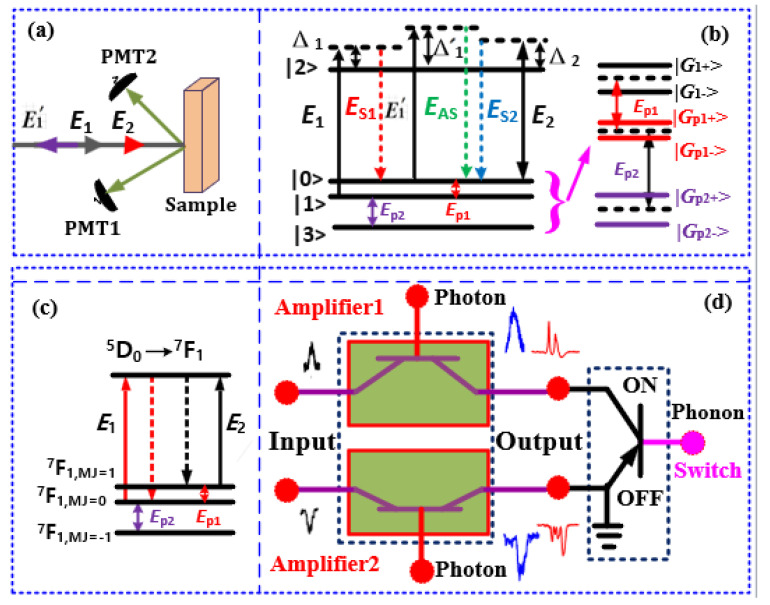
(**a**) Experimental setup, (**b**) Shows photon and phonon four-dressing energy level. (**c**) Shows energy levels of Eu^3+^:BiPO_4_ for transition ^7^F_1_→^5^D_0_. (**d**) The schematic diagram of proposed optical transistor as an amplifier and switch.

**Figure 2 nanomaterials-12-04304-f002:**
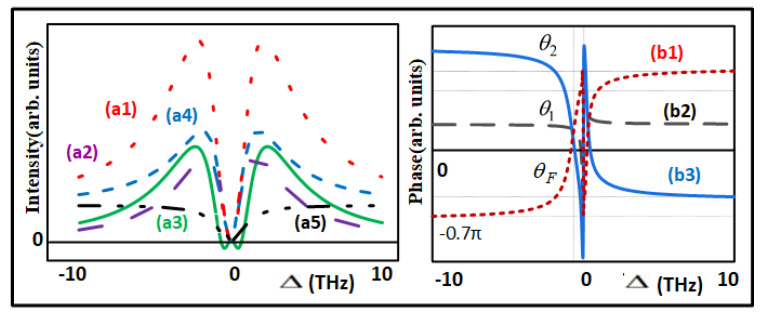
(**a1**) The total signal intensity of $|\rho_{F1}^{\left(2\right)}{|}^{2}+|\rho_{F2}^{\left(2\right)}{|}^{2}$ (hot curve), (**a2**) the interaction item $2|\rho_{F1}^{\left(2\right)}||\rho_{F2}^{\left(2\right)}|\cos(\theta )$ versus Δ (purple curve), (**a3**) $|\rho_{sum}^{\left(2\right)}{|}^{2}$ (green curve), (**a4**) $|\rho_{F1}^{\left(2\right)}{|}^{2}$ (blue curve), (**a5**)$|\rho_{F2}^{\left(2\right)}{|}^{2}$ (black curve). Figure 2b: The parameters are $G_{1}=2.3$ THz, $G_{2}=6.1$ THz. (**b1**) $\theta_{F}$ (hot curve), (**b2**) $\theta_{1}$ (black curve), (**b3**) $\theta_{2}$ (blue curve) versus Δ. Evolution of $\theta_{F}$, the constructive and the destructive interaction versus Δ. Figure 2b: The destructive or constructive interaction is studied in this system [23].

**Figure 3 nanomaterials-12-04304-f003:**
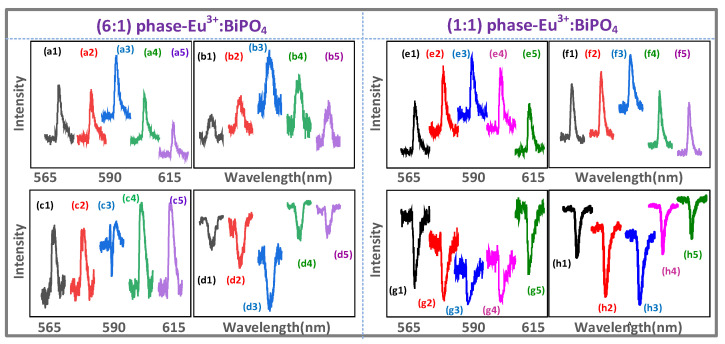
(**a**,**c**) show self- and cross- interaction of FL observed from Eu^3+^ doped BiPO_4_ [molar ratio (6:1)] at different ***E*_1_** wavelengths (567.4 nm, 584.4 nm, 587.4 nm, 589.4 nm, 612.4 nm) and ***E*_2_** scanned from 567.4 nm to 607.4 nm at PMT1 (far detector position) and PMT2 (near detector position), respectively. (**b**,**d**) show self- and cross- interaction of FL at different ***E*_2_** wavelengths (567.4 nm, 587.4 nm, 588 nm, 588.4 nm, 602.4 nm) and ***E*_1_** scanned from 567.4 nm to 612.4 nm. (**e**–**h**) show spectral signal intensity for (1:1) sample, which is same condition as (**a**–**d**). The time gate = 1 μs.

**Figure 4 nanomaterials-12-04304-f004:**
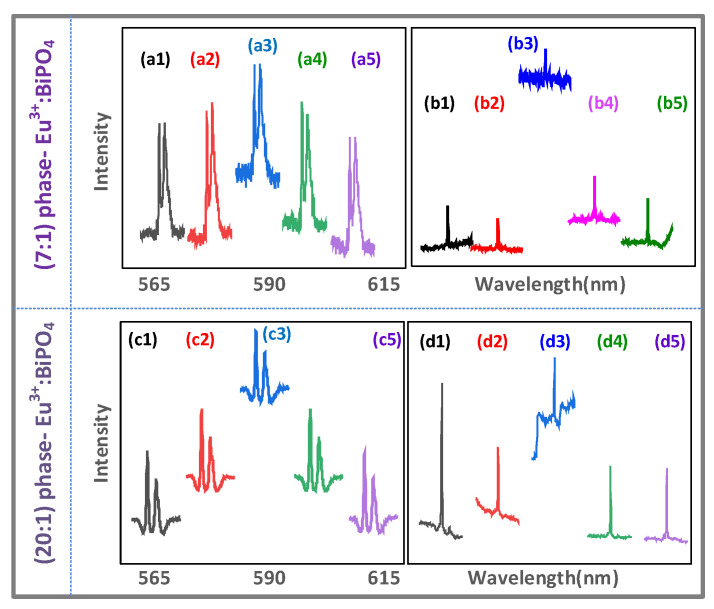
(**a1**–**a5**,**b1**–**b5**) show SFWM cross-interaction observed from Eu^3+^ doped BiPO_4_ [molar ratio (7:1)] at different narrowband laser ***E*_2_** (567.4 nm, 587.4 nm, 588 nm, 588.4 nm, 602.4 nm) while broadband laser ***E*_1_** is scanned from 572.4 nm to 612.4 nm at different broadband laser ***E*_1_** wavelengths (567.4 nm, 584.4 nm, 587.4 nm, 596.4 nm, 612.4 nm) and narrowband laser ***E*_2_** is scanned from 567.4 nm to 607.4 nm at 300 K, respectively, at PMT1. The time gates are 5 μs and 20 μs, respectively, gate width = 400 ns. (**c**,**d**) show SFWM cross-interaction for the (20:1) sample at the time gate = 10 μs and 20 μs, respectively. The other experimental condition is the same as (**a**,**b**), respectively, at PMT1.

**Figure 5 nanomaterials-12-04304-f005:**
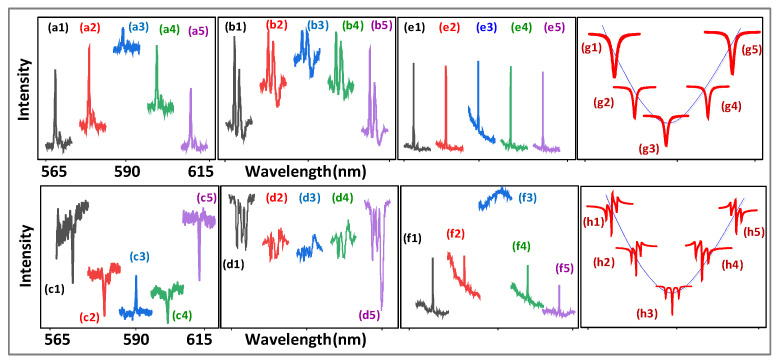
(**a**,**c**) show SFWM cross- interaction observed from Eu^3+^ doped in molar ratio (6:1) BiPO_4_ at different ***E*_1_** wavelengths (567.4 nm, 584.4 nm, 588.4 nm, 596.4 nm, 612.4 nm) and ***E*_2_** scanned from 567.4 nm to 607.4 nm at PMT1 and PMT2 at 300 K, respectively. (**b**,**d**) show SFWM cross- interaction at different ***E*_2_** wavelengths (567.4 nm, 587.4 nm, 588 nm, 588.4 nm, 602.4 nm) and ***E*_1_** scanned from 567.4 nm to 612.4 nm at PMT1 and PMT2 in 300 K, respectively. (**e**,**f**) show SFWM cross-interaction at 77 K. The other experimental conditions are the same as (**a**,**c**), respectively. Time gate = 500 μs. (**g1**–**g5**) show the simulation result corresponding to (**b1**–**b5**). (**h1**–**h5**) show the simulation result corresponding to Figure 3(g1–g5) and Figure 7(e1–e5).

**Figure 6 nanomaterials-12-04304-f006:**
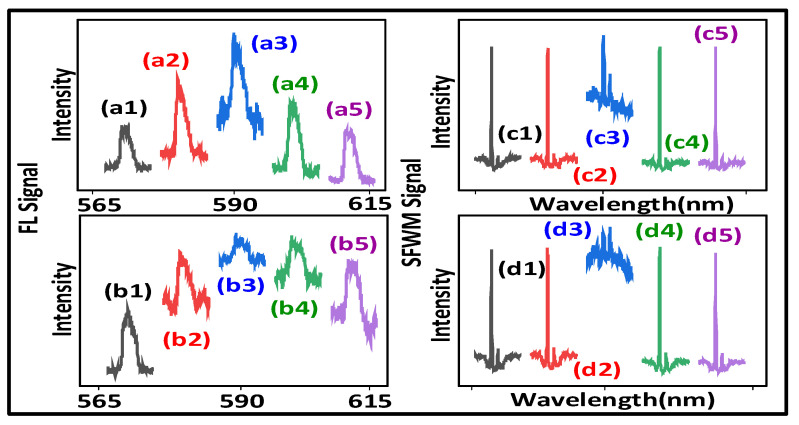
(**a**,**b**) show FL cross-interaction observed from the output signals of Eu^3+^ doped in molar ratio (0.5:1) BiPO_4_ at different ***E*_2_** wavelengths (567.4 nm, 587.4 nm, 588 nm, 588.4 nm, 602.4 nm) and ***E*_1_** scanned from 572.4 nm to 612.4 nm at 300 K, at PMT1 and PMT2, respectively. The time gate is 10 μs. (**c**,**d**) show SFWM cross-interaction at 77 K at the time gate = 800 μs, respectively. The other experimental condition is the same as (**a**,**b**).

**Figure 7 nanomaterials-12-04304-f007:**
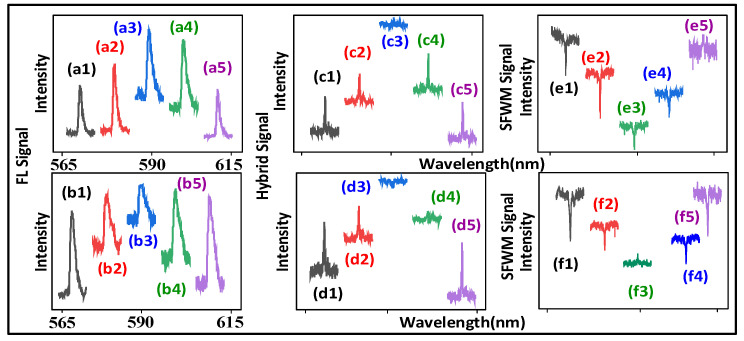
(**a**,**b**) show FL cross-interaction observed from Eu^3+^ doped in molar ratio (0.5:1) BiPO_4_ at different ***E*_1_** wavelengths (577.4 nm, 584.4 nm, 587.7 nm, 592.4 nm, 612.4 nm) and ***E*_2_** scanned from 567.4 nm to 607.4 nm at PMT1 and PMT2, respectively, at the near time gate (1 μs). (**c**,**d**) show hybrid cross-interaction at the middle time gates (100 μs). (**e**,**f**) show SFWM cross-interaction at the far time gate (500 μs). The other experimental condition is the same as (**a**,**b**), respectively.

**Table 1 nanomaterials-12-04304-t001:** Experimental parameters, Variables in the equations and Corresponding definition.

Experimental Parameters	Variables	Corresponding Definition
Time gate	$\rho_{F}^{\left(2\right)}/\rho_{AS}^{\left(3\right)}$	FL/SFWM density matrix
Temperature	$G_{i}/G_{pi}^{T}$	Photon Rabi frequency /phonon Rabi frequency
Sample	$\Delta_{i}/\Delta_{pi}^{j}$	Photon frequency detuning/phonon frequency detuning
Band excitation		
PMT	$\theta_{Fi}$	FL phase

**Table 2 nanomaterials-12-04304-t002:** Evolution of $\theta_{F}$, the constructive and the destructive interference versus Δ.

$\Delta =\Delta_{1}-\Delta_{2}$	$\left[-10{,}^{}-1.1\right)$	$\left[-1.1{,}^{}-0.2\right)$	$\left[-0.2{,}^{}0.2\right)$	$\left[0.2{,}^{}1.1\right)$	$[1.1{,}^{}10]$
$\theta_{F}=\theta_{1}-\theta_{2}$	[−0.7π,0.5π)	[0.5π,0.7π)	[−0.5π,0.5π)	[−0.7π,−0.5π)	[−0.5π,0.7π]
interaction	constructive	destructive	constructive	destructive	constructive

## Data Availability

Data underlying the results presented in this paper are not publicly available at this time but may be obtained from the authors upon reasonable request.

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
