# Peer review of "Photon–Phonon Atomic Coherence Interaction of Nonlinear Signals in Various Phase Transitions Eu^3+^: BiPO_4"

_nanomaterials, 2022, doi:10.3390/nano12234304_

Round 1

Reviewer 1 Report

Fan and coworkers investigated the photon-phonon atomic coherence interactions for various phase transitions in Eu3+:BiPOcrystals. Different atomic coherence spectral interactions can be probed by changing the time gates. The authors show that increased phonon dressing led to a destructive interaction of atomic coherences, while decreased dressing resulted in constructive interaction. Overall, this is an interesting, insightful, and relevant work for the readers of Nanomaterials. The authors also support their conclusions with adequate theoretical and experimental results. Therefore, I recommend the publication of this work in its present form.

Author Response

The authors would like to sincerely thank the respected reviewer for such valuable comments. 

Reviewer 2 Report

The authors have theoretically and experimentally investigated constructive and destructive photon-phonon atomic coherence interaction obtained from various phase transitions of Eu3+: BiPO4 crystal by changing time gates. The quality of work is excellent. The paper is scientifically sound. The results are original and clearly presented. Thus I recommend the paper to be published.

Author Response

The authors would cordially like to thank the respected reviewer for encouraging and supporting comments. Please see the attachment

Reviewer 3 Report

I read with much interest the manuscript by Fan et al, which deals with a topic that is certainly of interest to specialists in this field. However, I think the choice of Nanomaterials as a journal to disseminate these results is not the most appropriate. I think the manuscript should be submitted to a more specialized journal where it can surely be appreciated.

However, some observations can certainly be made.The first concerns an attempt to adopt a more compact and , in some parts more concise symbology, so as to make reading the manuscript easier even for non-super-specialist readers.

Figures 3, 4 and 5 are of poor quality and it would be good to standardize the scale of the abscissae.

It seems to me that there are incorrectly written symbols at lines 174 et seq. Please check.

The conclusions are too narrow and should be expanded to better illustrate the results presented in the manuscript

Reviewer 4 Report

The paper presents a detailed experimental and theoretical study of photon-phonon atomic coherence interaction in Eu3+:BiPO4. The dynamics of this atomic coherence interaction is used to construct an optical transistor functioning as an amplifier or switch. The article is written well and is structured logically.  The paper can be considered for publication in Nanomaterials pending the following revisions on the manuscript:

1. The theoretical model in section 2.1 is highly detailed however, it would be very daunting for the general reader to understand clearly the equations and expressions the way they are presented.  Therefore it will benefit the readers if at the end of Section 2.1 that a table be constructed with one column listing the variables and parameters used in the equations, and on the second column the corresponding definition or descriptive name of the the variable or parameter.

2. It would be highly insightful if the authors would correlate which parts of Figure 1(d) corresponds to Figure 1(c).  It doesn’t have to be a new figure but can be described within the text of Section 2. 

3. The conclusion is too short.  It should include not just the general outcome of the experimental results and the comparison with the theoretical model. The conclusion should also include some quantitative key results instead of just qualitative descriptions of the results.

After these revisions have been added to the paper, it could be considered for publication in Nanomaterials Journal.

Round 2

Reviewer 3 Report

The authors responded satisfactorily to the comments and observations of my previous report, and in particular I appreciated the changes in the symbology and the improved version of the figures. Therefore, the work can be published in the present version.